# Speech Separation Using an Asynchronous Fully Recurrent Convolutional Neural Network

**Xiaolin Hu**[1]*, **Kai Li**[1], **Weiyi Zhang**[1], **Yi Luo**[2], **Jean-Marie Lemercier**[3], **Timo Gerkmann**[3]

[1]Department of Computer Science and Technology,
Tsinghua Laboratory of Brain and Intelligence (THBI),
IDG/McGovern Institute of Brain Research Tsinghua University, Beijing, China
[2]Department of Electrical Engineering, Columbia University, NY, USA
[3]Department of Informatics, University of Hamburg, Hamburg, Germany
xlhu@tsinghua.edu.cn, {lk21,wy-zhang19}@mails.tsinghua.edu.cn,
y.luo@columbia.edu, lemercier@informatik.uni-hamburg.de,
timo.gerkmann@uni-hamburg.de

## Abstract

Recent advances in the design of neural network architectures, in particular those specialized in modeling sequences, have provided significant improvements in speech separation performance. In this work, we propose to use a bio-inspired architecture called Fully Recurrent Convolutional Neural Network (FRCNN) to solve the separation task. This model contains *bottom-up*, *top-down* and *lateral* connections to fuse information processed at various time-scales represented by *stages*. In contrast to the traditional approach updating stages in parallel, we propose to first update the stages one by one in the bottom-up direction, then fuse information from adjacent stages simultaneously and finally fuse information from all stages to the bottom stage together. Experiments showed that this asynchronous updating scheme achieved significantly better results with much fewer parameters than the traditional synchronous updating scheme. In addition, the proposed model achieved good balance between speech separation accuracy and computational efficiency as compared to other state-of-the-art models on three benchmark datasets.

## 1 Introduction

Speech separation aims to extract individual speeches from a mixture of speeches of multiple speakers. It is an important preprocessing step for speech recognition in noisy environment. Recent development of speech separation methods at the waveform level has aroused researchers' interest [21, 22, 20], avoiding the traditional representation of STFT amplitude and phase used in so-called time-frequency (TF) domain methods [7, 11]. Among these so-called time-domain methods, some presented mechanisms fuse information processed at various time scales, called *multi-scale fusion (MSF)* methods, such as in FurcaNeXt [40] or SuDoRM-RF [33], and yield impressive results on the speech separation task. In this work we aim to explore if there exist even better MSF methods.

Evidence from observations of sensory systems of mammals show them to utilize MSF in their processing. For instance, the visual system includes multiple processing stages (from lower functional areas such as the lateral geniculate nucleus to higher functional areas such as the inferior temporal cortex), which process different scales of information [1]: the higher the stage, the coarser the scale. See Figure 1a for illustration. Similar mechanisms and areas have also been identified and located in the auditory system [1]. More importantly, physiological and anatomical studies have revealed abundant recurrent synaptic connections within the same stage (also called *lateral*

---

*Corresponding author.

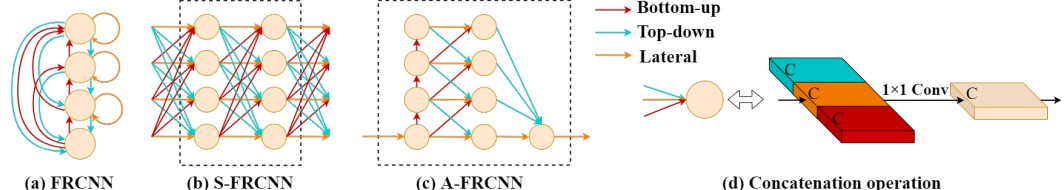

Figure 1: The structure of FRCNN and typical updating schemes. The number of stages $S = 4$. (a) The structure of the FRCNN. Every node denotes a stage, corresponding to a group of neurons in a functional area in the sensory pathway (e.g., the inferior colliculus in the auditory pathway). Red, blue and orange arrows denote bottom-up, top-down and lateral connections, respectively. Both bottom-up and top-down connections can be made between adjacent stages and non-adjacent stages. (b) Synchronous updating scheme in one block [19]. (c) The proposed asynchronous updating scheme in one block. The dashed box in each subfigure indicates the basic building block for constructing a complete RNN (see Figure 4). (d) Multi-scale information fusion for an example stage receiving three types of inputs.

*connections*) and bottom-up/top-down synaptic connections between stages [3]. The intra-stage and inter-stage connections bring different scales of sensory information together and each stage performs information fusion. These connections fuse different scales of information more completely, and may lead to better results than existing MSF methods.

However, Figure 1a merely reflects a purely static structure in the brain and does not show the dynamics of the sensory system. In biological systems, given a stimulus, the neurons along a sensory hierarchy do not fire simultaneously like shown in Figure 1b. For example, it was reported that the neural response initialized at a retinotopic position in anesthetized rat V1 propagated uniformly in all directions with a velocity of 50–70 mm/s, slowed down at the V1/V2 area border, after a short interval, spread in V2, then reflected back in V1 [38]. In general, "the speed of an action potential varies among neurons in a range from about 2 to 200 miles per hour"[24]. The time at which a neuron starts to fire depends on a variety of factors including the neuron type, the stage at the sensory pathway, the number of the dendrites connected to it and the morphology of the neural fibers. This precludes the possibility of faithfully replicating the sensory system to obtain an excellent artificial neural network (ANN). Nevertheless, the history of ANN development indicates that getting inspiration from the brain is enough to make great progress if task-specific techniques are combined. Inspired by the discovery of simple cells and complex cells in cat visual cortex [9, 10], a hierarchical model Neocognitron [6] was proposed and later developed into convolutional neural networks [15] by applying the backpropagation algorithm. We investigate empirically if there exists an asynchronous updating scheme for the structure shown in Figure 1a that provides improvement for speech separation performance.

As the model has bottom-up, top-down and lateral connections as shown in Figure 1a, we call the model a *fully recurrent convolutional neural network (FRCNN)*. This name emphasizes the presence of both lateral and top-down recurrent connections in the model, distinguishing the model from an existing model [17] named *recurrent convolutional neural network (RCNN)* that has lateral recurrent connections only. The model with the synchronous updating scheme (Figure 1b) is called the synchronous FRCNN or S-FRCNN, which was studied for visual recognition [19]. We aim to propose an asynchronous FRCNN or A-FRCNN for speech separation. We notice that SuDoRM-RF [33] also has the three types of connections and we start from its framework to study different updating schemes of FRCNN.

The architecture of our proposed A-FRCNN is illustrated in Figure 1c. The information first passes through stages one by one in the bottom-up direction, then fuses between adjacent stages in parallel, and finally fuses together with skip connections to the bottom stage. In the S-FRCNN, the information transmission from the bottom stage to any upper stage then back to the bottom stage is too fast: one step upward and one step downward (Figure 1b). In contrast, in the A-FRCNN, the information starting from the bottom stage goes through more processing steps before it goes back to the bottom stage, which is advantageous for comprehensive MSF. Increasing the depth of a model is one of the keys for the success of deep learning. We will show the merit of A-FRCNN compared to S-FRCNN in experiments.

## 2 Related Work

### 2.1 Speech Separation Methods

A typical speech separation method is to model different sources in the temporal-frequency (TF) domain. First, the short-time Fourier transform (STFT) calculates the TF representation of the mixed sound. Second, the subsequent process approximates the clean spectrogram of each source from the mixed spectrogram and uses the inverse STFT (iSTFT) to synthesize the source waveform, as in DPCL [7], uPIT [12] etc.

So-called time-domain methods were also proposed for speech separation, making use of non-STFT encoders for extracting meaningful representations out of the waveform (in a bio-inspired fashion [4], or fully learned over training [22]). DualPathRNN [20] extracts overlapped short sequences (called *chunks*) from the mixed speech signal and applies intra- and inter-chunk operations iteratively by using recurrent neural networks (RNNs). It has achieved very good results at the cost of high computational complexity. The idea of intra-chunk and inter-chunk was adopted in recently proposed models such as Sepformer [31], Sandglasset [13] and Gated DualPathRNN [23]. These models achieved even better speech separation results but the computational complexity is also higher.

While RNNs were naturally used to perform speech separation given the sequential aspect of the input representation, they often require long training and inference time. Conv-TasNet [22] has been proposed to solve this problem, replacing RNN with Temporal Convolutional Networks, much faster to train. However, this model has limited MSF capability. Recently, several models with multiple branches have been proposed for speech separation, where different branches adopt different time resolutions for the processing of their respective inputs, then the outputs are fused with some rule or dedicated module. For instance, SuDoRF-RF [33] uses repetitive U-Nets [28], obtaining good results with high efficiency. FurcaNeXt [40], a variant of Conv-TasNet [22], uses multiple branch learning methods to improve the performance of speech separation. MSGT-TasNet [41] uses Transformer [34] to capture features of different scales for speech separation.

### 2.2 Lateral and Top-Down Recurrent Connections

The lateral and top-down recurrent connections have been modeled by ANN researchers for a long time. In 1990s recurrent connections were introduced into the multi-layer Perceptrons [5] [27], and in 2015 they were introduced into the CNN, resulting in the RCNN [17][18]. In 2000s top-down connections were introduced into unsupervised deep learning models [8] [16]. A general framework with both lateral and top-down recurrent connections was proposed in 2016 [19]. If a hierarchical model has recurrent connections, the neurons can be updated in different orders. In [19], only the conventional synchronous updating scheme was presented. However, no evidence has shown that this is how the neural system works, or that it outperforms asynchronous updating schemes on engineering tasks. In fact, in [18], it was shown that an asynchronous updating scheme for RCNN outperformed the synchronous updating scheme on an image segmentation task. We here propose a novel asynchronous scheme for the FRCNN that achieved better results with fewer parameters than the synchronous scheme on speech separation.

## 3 Methods

### 3.1 Overall pipeline

An algorithm dedicated to speech separation aims to extract individual speech signals of different speakers from a mixture. We denote the waveform of the mixture as $\mathbf{x} \in R^{1 \times T}$:

$$\mathbf{x} = \sum_{i=1}^{C} \mathbf{s}_i + \sigma \tag{1}$$

where $\mathbf{s}_i \in R^{1 \times T}$ denotes the waveform of speaker $i$, $\sigma \in R^{1 \times T}$ denotes the noise signal, $T$ denotes the number of samples of the signal, and $C$ denotes the number of speakers. The task is to estimate $\mathbf{s}_i$ from $\mathbf{x}$ for all $i$.

We use the same pipeline as Conv-TasNet [22], as shown in Figure 2. It consists of an encoder, a separation network and a decoder. The encoder divides $\mathbf{x}$ into $K$ overlapping segments $\overline{\mathbf{x}}_k \in R^{1 \times L}$ and transforms each segment into a feature vector $\overline{\mathbf{r}}_k \in R^{1 \times N}$:

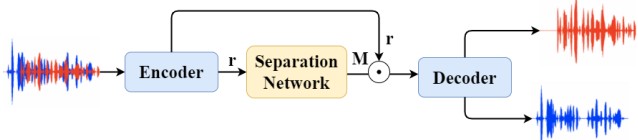

Figure 2: The overall pipeline for speech separation.

$$\overline{\mathbf{r}}_k = \overline{\mathbf{x}}_k \mathbf{U}_e \tag{2}$$

where $\mathbf{U}_e \in R^{L \times N}$ is a weight matrix. The two steps can be realized by a trainable 1D convolution with kernel $\mathbf{U}_e$ and an appropriate stride.

The separation network receives $\overline{\mathbf{r}}_k$ to estimate a mask $\mathbf{M}_i \in R^{1 \times N}$ for speaker $i$. We apply a fully-connected layer with ReLU activation to the output of the separation network to produce $\mathbf{M}_i$. The detailed structure of the separation network is introduced in Section 3.2.

The decoder reconstructs the waveform segment

$$\overline{\mathbf{s}}_{i,k} = (\overline{\mathbf{r}}_k \odot \mathbf{M}_i) \mathbf{U}_d^\top \tag{3}$$

where $\mathbf{U}_d \in R^{L \times N}$ is a weight matrix and $\mathbf{U}_d^\top$ is the transpose of $\mathbf{U}_d$, and $\odot$ stands for element-wise multiplication. The estimated waveform $\widehat{\mathbf{s}}_i$ is obtained by summing $K$ overlapping segments $\overline{\mathbf{s}}_{i,k}$. The two steps can be realized by a 1-D transposed convolution operation.

## 3.2  Separation Network

### 3.2.1  Structure of FRCNN

We use the FRCNN as the separation network. It can be represented by a graph with nodes denoting stages and edges denoting connections. Figure 1a shows an example with $S = 4$ stages. In biological terms, every node corresponds to a set of neurons in a certain stage in the sensory pathway, e.g., the inferior colliculus in the auditory pathway. In our model, every node corresponds to a convolutional layer. Different nodes process different scales of the input information. The higher the node, the coarser the information. There are three types of connections: bottom-up, top-down and lateral connections. Note that both bottom-up and top-down connections can be between adjacent stages and non-adjacent stages. In the latter case, the connections are called *skip-connections*.

### 3.2.2  Updating Schemes in the Micro-level

To run a recurrent neural network (RNN) with intricate connections, one needs to first determine the updating order of the neurons. This order determines the RNN *unfolding* or *unrolling* scheme. A commonly used approach is to update all neurons simultaneously. In the case of FRCNN as shown in Figure 1a, it corresponds to updating all stages synchronously. This scheme is depicted in Figure 1b [19], and denoted by S-FRCNN. However, if the stages are allowed to be updated asynchronously, there will be a large number of possible unfolding schemes. For example, without considering the skip connections, we can update the stages one by one in the upward direction then update them one by one in the downward direction. In the present work, we propose an efficient updating scheme A-FRCNN, as shown in Figure 1c.

In the proposed A-FRCNN, we first sequentially update the stages in the bottom-up direction, then update them simultaneously by fusing information from adjacent stages, and finally, fuse information from all stages to the bottom stage. This entire process is repeated several times (Sec. 3.2.4). The two types of fusions can be viewed as local and global fusions, respectively. As a stage represents a unique set of neurons as its biological counterpart, the connections between two stages (e.g., the vertical upward connection and the oblique upward connection between stage 3 and stage 4) should use the same operation and parameters.

The A-FRCNN is adapted from the S-FRCNN in a step-by-step manner.

1. Inspired by the structure of the U-Net [28], we design a bottleneck structure at the bottom stage as shown in Figure 3a. All upper stages exchange information between different blocks through the bottom stage. This design increases the number of steps including

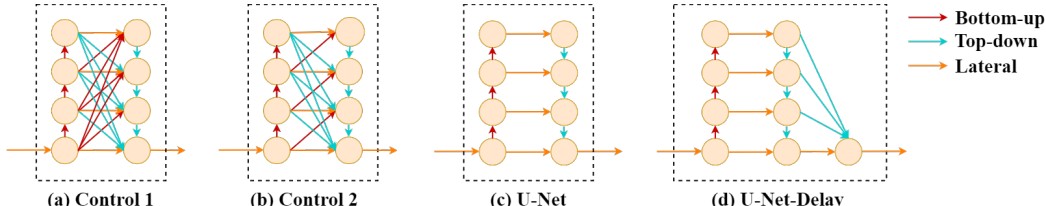

Figure 3: Various asynchronous updating schemes in the micro-level.

down-sampling and up-sampling operations for processing the input coming at the highest resolution. This block is denoted by Control 1.

2. The block Control 1 has too many connections which make the model inefficient in both parameters and computation. The bottom-up skip-connections and top-down skip-connections are symmetric, and may be redundant. Therefore, we remove the bottom-up skip-connections, which results in the block shown in Figure 3b, denoted by Control 2.

3. It is too fast to fuse the information across non-adjacent stages through top-down skip-connections in the block Control 2. One possible way to represent an increasing firing delay from widely separated units would be to fuse the information across adjacent stages first, then across non-adjacent stages. This change increases the shortest path from higher stages to the bottom stage. In addition, to save parameters and computation, we only keep the top-down skip-connections to the bottom stage and removed other top-down skip-connections. We also remove the vertical downward connections because the top-down stage-by-stage fusion has already been performed through the oblique downward connections. This is made possible by the delayed global fusion; otherwise, the stages would become disconnected after removing the vertical downward connections. We then obtain the A-FRCNN (Figure 1c).

Note that the sequential fusion method in the third step is more biologically plausible than the synchronous fusion method since biological connections between non-adjacent stages are longer than those between adjacent stages, while signal transmission through connections is not instantaneous.

The proposed A-FRCNN block is closely related to the U-Net [28] (Figure 3c). To investigate the potential advantage of delayed global fusion, we add top-down skip-connections to the U-Net block and obtain a new block, denoted by U-Net-Delay (Figure 3d).

### 3.2.3 Multi-scale Information Fusion inside Blocks

The blocks depicted in Figures 1 and 3 are RNN blocks, and the nodes in the same horizontal row represent the same stage (or in biological terms, the same set of neurons in a sensory area) but at different time. In this study we use $C$ feature maps for every stage. Multi-scale information fusion is performed at the input of every stage. First the $C$ feature maps from each of the $K$ inputs are concatenated in the channel dimension, resulting in $KC$ feature maps. A $1 \times 1$ convolutional layer is then used to reduce the number of feature maps to $C$. Figure 1d illustrates this process. This concatenation method was used by default in our experiments. One can also sum up the $K$ inputs to obtain $C$ feature maps.

### 3.2.4 Unfolding Methods in the Macro-level

Figures 1 and 3 show single blocks of the entire unfolding schemes. An entire unfolding scheme usually consists of multiple such blocks with tied weights. If there are $B$ blocks in total, we say "FRCNN is unfolded for $B$ time steps". At the macro-level, the FRCNN can be unfolded by simply repeating these blocks along time such that the output of one block is the input of the next block.

To further fuse the multi-scale information, we add a $1 \times 1$ convolution between two consecutive blocks (Figure 4a). This method is formulated as follows:

$$R(t + 1) = f(\varphi(R(t))), \tag{4}$$

where $f(\cdot)$ denotes a block shown in Figures 1 and 3, $R(t)$ denotes the output of the block at time step $t$ and $\varphi$ denotes $1 \times 1$ convolution. This is called the *direct connection (DC)* method.

Another idea is to integrate the input of the model with the output of every block via feature map concatenation or summation before sending to the next block. This rule was used in constructing the recurrent CNN in a previous study [17]. Again, we add a $1 \times 1$ convolution to further fuse information (Figure 4b). Formally,

$$R(t + 1) = f(\varphi(R(t) \oplus \mathbf{r})) \quad (5)$$

where $\mathbf{r}$ denotes the input feature maps and $\oplus$ denotes concatenation or summation of two sets of feature maps. This is called the *concatenation connection (CC)* or *summation connection (SC)* depending on which feature map integration method is used.

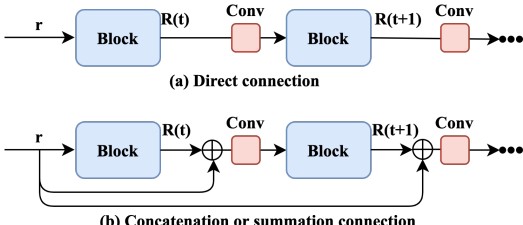

Figure 4: Macro-level unfolding schemes of the FR-CNN. Every blue box corresponds to a dashed box in Figures 1 and 3. The pink boxes in a model denote $1 \times 1$ convolutions with shared weights.

For single-input-single-output blocks, i.e., A-FRCNN and the blocks shown in Figure 3, we directly use the unfolding methods as described above. For the multi-input-multi-output block, i.e., S-FRCNN, we apply these unfolding methods for each input-output pair corresponding to the same stage. It should be noted that Figure 1b only illustrate the intermediate blocks of S-FRCNN unfolding scheme. In the beginning of unfolding we use downsampling to obtain different scales of feature maps, and in the end of unfolding we use up-sampling to fuse different scales of feature maps.

### 3.3 Training Method

We use the standard BP algorithm to train the model. The object is to maximize the scale-invariant signal-to-noise ratio (SI-SNR) [14]. See Supplementary Materials for details. SI-SNR is also a metric to evaluate the performance of speech separation methods.

## 4 Experimental Settings

### 4.1 Dataset

**Libri2Mix** [2]. This dataset was constructed using train-100, train-360, dev, and test set in the LibriSpeech dataset [25]. Random extracts were selected for different speakers and mixed with uniformly sampled Loudness Units relative to Full Scale (LUFS) [29] between -25 and -33 dB. Random noise samples were added, with loudness uniformly sampled between -38 and -30 LUFS. We used train-100 as the training set which has 58 hours. We used the same test set to compare different methods.

**WSJ0-2Mix** [7]. This dataset contains a 30-hour training set, a 10-hour validation set and a 5-hour test set. It was generated by combining the speech signals of different speakers in the Wall Street Journal corpus. The speech signals were randomly selected and mixed with a Signal-to-Noise Ratio uniformy sampled between -5dB and 5dB.

**WHAM!** [37]. WHAM! added noise to WSJ0-2Mix, which was recorded in scenes such as cafes, restaurants and bars. The speeches were mixed with noise with a SNR uniformly sampled between -6dB and 3dB. Due to the existence of noise, WHAM! is more challenging for speech separation than WSJ0-2Mix.

### 4.2 Implementation Details

The encoder and decoder in the speech separation pipeline were respectively a 1-D convolutional layer and a 1-D transposed convolutional layer. We set their kernel size to 21, stride to 10, and channel number to 512, i.e., $L = 21$ and $N = 512$. Unless otherwise specified, the number of stages $S$ was set to 5, the number of channels $C$ in every stage was set to 512 and the SC method was used for unfolding in the macro-level. Whenever $C \neq N$, a $1 \times 1$ convolutional layer was added between the encoder and the separation network.

We designed two methods to realize the connections shown in Figures 1 and 3.

- Method A: The bottom-up and top-down connections were realized by convolution (kernel size 5 and stride 2) and the PixelShuffle technique [30] (kernel size 5), respectively. The PixelShuffle technique was shown to be better than other upsampling techniques for image super-resolution reconstruction. The lateral connections were realized by $1\times1$ convolution.

- Method B: The bottom-up connections were realized by the convolution operation with kernel size 5 and appropriate strides. For example, one operation was used for $2\times$ down-sampling and two consecutive operations were used for $4\times$ down-sampling, and so on. The top-down connections were realized by interpolation. The lateral connections were realized by simply copying the feature maps.

All convolutions were depthwise separable convolutions. In Method A all connections had trainable parameters, resembling plastic synapses in biological systems. In Method B, only the bottom-up connections had parameters, and it is therefore less biologically plausible. However, Method B is more parameter efficient and computing efficient.

We trained all models for 200 epochs on 3-second utterances for Libri2Mix and 4-second utterances for WHAM! and WSJ0-2Mix with 8K Hz sampling rate. Batch size was set to 8. The initial learning rate of Adam optimizer was $1 \times 10^{-3}$, and it decayed to $1/3$ of the previous rate every 40 epochs. During training, gradient clipping with a maximum $l_2$-norm of 5 was used. For each architecture, we picked the best model based on its results on the validation set, then performed testing.

All experiments were conducted on a server with Intel(R) Xeon(R) Silver 4210 CPU @ 2.20GHz and GeForce RTX 1080 Ti 11G $\times$8. The Pytorch implementation of the models is publicly available[†]. It is based on the code of SuDoRM-RF[‡]. This project is MIT Licensed.

# 5    Results

We used the scale-invariant signal-to-noise ratio improvement (SI-SNRi) [14] and signal-to-distortion ratio improvement (SDRi) [35] as the evaluation metrics to measure the speech separation accuracies of models. Their definitions are found in Supplementary Materials. In Secs. 5.1 and 5.2, we report the mean±std of these metrics over 5 models trained from different random seeds. We report inference time on CPU, indicated by "Time" in tables throughout the paper. It was calculated by processing a four-second audio on CPU then averaged over 1000 trials.

## 5.1    Comparison of Micro-level Updating Schemes

We compared the performances of the micro-level updating schemes on Libri2Mix. The results are presented in Table 1. By using either Method A or Method B for implementing the connections (Sec. 4.2), the proposed A-FRCNN scheme performed better than the S-FRCNN scheme and the two control schemes. By comparing the results of S-FRCNN and Control 1, we see that the bottleneck design brought a large improvement on the SI-SNRi and SDRi metrics. Control 1 and Control 2 achieved similar SI-SNRi and SDRi values indicating that the bottom-up skip-connections are indeed redundant. The delayed global fusion further improved the results of Control 2. From S-FRCNN to Controls 1 and 2 and A-FRCNN, the number of connections decreased; as a result, the amount of parameters and inference time also decreased.

For implementing the connections in the four models in Table 1, Method B obtained similar or even better results with fewer parameters and less inference time than Method A. Therefore, we adopted Method B in later experiments. All results reported in what follows were obtained with Method B.

The S-FRCNN had much more parameters than the A-FRCNN. We then reduced $C$ to 412 and obtained a new model, S-FRCNN (light), which had similar number of parameters to the A-FRCNN. Its results were even worse than the original S-FRCNN (Table 1), suggesting that the poor performance of the S-FRCNN was not due to overfitting caused by large amount of parameters. We removed all skip-connections from S-FRCNN and obtained a model called S-FRCNN (no-skip). A CNN model with similar architecture has been used in face parsing [42, 39], but different blocks do not share

---

[†] `https://cslikai.cn/project/AFRCNN`
[‡] `https://github.com/etzinis/sudo_rm_rf/blob/master/sudo_rm_rf/dnn/models/improved_sudormrf.py`

Table 1: Comparison of different FRCNN updating schemes on the Libri2Mix test set. The values are presented as "Method A / Method B". Each model was unfolded for 8 time steps.

| Model | SI-SNRi | SDRi | Params (M) | Time (s) |
|---|---|---|---|---|
| S-FRCNN | $12.6_{\pm 0.05}$ / $12.1_{\pm 0.05}$ | $13.0_{\pm 0.10}$ / $12.5_{\pm 0.05}$ | 9.8/ 9.6 | 1.35 / 1.12 |
| Control 1 | $15.0_{\pm 0.06}$ / $14.6_{\pm 0.04}$ | $15.5_{\pm 0.06}$ / $15.0_{\pm 0.03}$ | 8.3 / 8.0 | 1.41 / 0.96 |
| Control 2 | $15.1_{\pm 0.01}$ / $14.6_{\pm 0.04}$ | $15.4_{\pm 0.02}$ / $15.0_{\pm 0.04}$ | 6.6 / 6.4 | 1.16 / 0.81 |
| A-FRCNN (ours) | $\mathbf{15.2}_{\pm 0.04}$ / $\mathbf{15.5}_{\pm 0.04}$ | $\mathbf{15.7}_{\pm 0.04}$ / $\mathbf{15.9}_{\pm 0.04}$ | 6.2 / 6.1 | 1.07 / 0.72 |
| S-FRCNN (light) | - / $11.8_{\pm 0.03}$ | - / $12.2_{\pm 0.02}$ | - / 6.4 | - / 0.97 |
| S-FRCNN (no-skip) | - / $11.43_{\pm 0.01}$ | - / $11.85_{\pm 0.02}$ | - / 6.4 | - / 0.51 |

Table 2: Comparison with the U-Net and U-Net-Delay on the Libri2Mix test set. Each model was unfolded for 8 time steps.

| Model | SI-SNRi | SDRi | Params (M) | Time (s) |
|---|---|---|---|---|
| U-Net | $11.4_{\pm 0.01}$ | $11.8_{\pm 0.01}$ | 4.0 | 0.39 |
| U-Net-Delay | $13.2_{\pm 0.05}$ | $13.6_{\pm 0.05}$ | 5.1 | 0.60 |
| U-Net (large) | $11.5_{\pm 0.01}$ | $11.9_{\pm 0.01}$ | 6.1 | 0.69 |
| U-Net-Delay (large) | $12.1_{\pm 0.01}$ | $12.5_{\pm 0.01}$ | 6.4 | 0.88 |
| A-FRCNN (ours) | $\mathbf{15.5}_{\pm 0.04}$ | $\mathbf{15.9}_{\pm 0.04}$ | 6.1 | 0.72 |

weights. From Table 1, it is seen that S-FRCNN (no-skip) achieved worse results than the original S-FRCNN.

The U-Net structure did not yield good results (Table 2). With delayed global fusion, the U-Net-Delay block yielded better results. For a fair comparison with the A-FRCNN, we changed the number of channels $C$ from 512 to 684 and 580 in U-Net and U-Net-Delay blocks, respectively, and obtained the U-Net (large) and U-Net-Delay (large) blocks. The two blocks had similar amount of parameters to the A-FRCNN. With more parameters, the U-Net-Delay (large) tended to overfit the training data and yielded worse test results than the original U-Net-Delay. Even in this case, U-Net-Delay (large) yielded better results than the U-Net (large), verifying the effectiveness of the delayed global fusion. Nevertheless, they both yielded much lower SI-SNRi and SDRi values than the A-FRCNN.

## 5.2 Comparison with Existing Models

We compared A-FRCNN with some popular models for speech separation. Some models work in the time-frequency domain: DPCL++ [7], uPIT-BLSTM-ST [12] and Chimera++ [36]. Some models work in the time-domain: BLSTM-TasNet [21], Conv-TasNet [22], Two-Step TDCN [32], MSGT-TasNet [41], SuDoRM-RF [33], DualPathRNN [20], Sepformer [31] and Gated DualPathRNN [23]. SuDoRM-RF has four variants which are labeled by appending 0.25x, 0.5x, 1.0x and 2.5x to the end of the name, indicating the variants consist of 4, 8, 16 and 40 blocks, respectively. None of these models have reported results on all of the three datasets and we run some of them to obtain missing results by using the Asteroid toolkit [26]. See Table 3. Sepformer and Gated DualPathRNN have achieved the best results on WSJ0-2Mix, but we could not afford the computational resource to obtain the missing results on other datasets as the two models require single GPU memory larger than 20 GB.

We tested three variants of A-FRCNN by unfolding for 4, 8, 16 times in the macro-level. We also tested variants in which the concatenation in Figure 1d was replaced with summation, and their names have "sum" attached to the end in Table 3.

**Separation accuracy.** On Libri2Mix we had the following observations. First, among existing models DualPathRNN achieved the best results. Second, with more unfolding times in the macro-level, A-FRCNN achieved better results at the cost of increasing inference time. Third, A-FRCNN-16 achieved better results than DualPathRNN.

Some audio examples from Libri2Mix separated by different models including DualPathRNN and A-FRCNN-16 are provided in Supplementary Materials. In most examples we found that separated speeches by A-FRCNN-16 sounded clearer than those by other methods.

On WHAM! and WSJ0-2Mix, DualPathRNN, Gated DualPathRNN and Sepformer obtained higher SI-SNRi and SDRi values than A-FRCNN-16, but we will show below that this is at the cost of

Table 3: Performance of different models. The SI-SNRi and SDRi values marked with '*' indicate that they were not reported in the original papers but obtained by using the Asteroid toolkit [26]. The toolkit is licensed under the MIT licence.

| Model | Libri2Mix | | WHAM! | | WSJ0-2Mix | | Params (M) | Time (s) |
|---|---|---|---|---|---|---|---|---|
| | SI-SNRi | SDRi | SI-SNRi | SDRi | SI-SNRi | SDRi | | |
| DPCL++ | 5.9* | 6.6* | - | - | 10.8 | 11.2 | 13.6 | 0.37 |
| uPIT-BLSTM-ST | 7.6* | 8.2* | - | - | 9.8 | 10.0 | 92.7 | 0.77 |
| Chimera++ | 6.3* | 7.0* | 10.0 | - | 11.5 | 11.8 | 32.9 | 0.66 |
| BLSTM-TasNet | 7.9* | 8.7* | 9.8 | - | 13.2 | 13.6 | 23.6 | 2.90 |
| Conv-TasNet | 12.2 | 12.7 | 12.7 | - | 15.3 | 15.6 | 5.6 | 0.39 |
| Two-Step TDCN | 12.0* | 12.5* | - | - | 16.1 | - | 8.6 | 1.85 |
| MSGT-TasNet (light) | - | - | 12.3 | - | 16.8 | 17.1 | 37.4 | 3.37 |
| MSGT-TasNet (dense) | - | - | 13.1 | - | 17.0 | 17.3 | 66.8 | 5.56 |
| SuDoRM-RF 0.25x | 10.8* | 11.3* | 10.4* | 10.8* | 13.4 | 13.6 | 0.8 | 0.24 |
| SuDoRM-RF 0.5x | 12.2* | 12.6* | 11.8* | 12.2* | 15.4 | 15.6 | 1.4 | 0.33 |
| SuDoRM-RF 1.0x | 13.5* | 14.0* | 12.9* | 13.3* | 17.1 | 17.3 | 2.7 | 0.53 |
| SuDoRM-RF 2.5x | 14.0* | 14.4* | 13.7* | **14.1*** | 17.4* | 17.6* | 6.4 | 1.15 |
| DualPathRNN | **14.1*** | **14.6*** | 13.7* | **14.1*** | 18.8 | 19.0 | 2.7 | 4.66 |
| Gated DualPathRNN | - | - | **15.2** | - | 20.1 | - | 7.5 | 7.35 |
| Sepformer | - | - | - | - | **20.4** | **20.5** | 26.0 | 5.22 |
| A-FRCNN-4 | 12.6 ±0.01 | 13.1 ±0.02 | 12.0 ±0.02 | 12.3 ±0.02 | 15.6 ±0.02 | 15.8 ±0.01 | 6.1 | 0.33 |
| A-FRCNN-8 | 15.5 ±0.04 | 15.9 ±0.04 | 13.4 ±0.02 | 13.8 ±0.02 | 17.1 ±0.03 | 17.3 ±0.02 | 6.1 | 0.72 |
| A-FRCNN-16 | **16.7** ±0.03 | **17.2** ±0.03 | **14.5** ±0.02 | **14.8** ±0.03 | **18.3** ±0.02 | **18.6** ±0.02 | 6.1 | 1.51 |
| A-FRCNN-4 (sum) | 13.1 ±0.01 | 13.5 ±0.02 | 12.3 ±0.02 | 12.6 ±0.02 | 14.9 ±0.02 | 15.2 ±0.02 | 1.7 | 0.29 |
| A-FRCNN-8 (sum) | 15.0 ±0.01 | 15.4 ±0.02 | 13.0 ±0.02 | 13.4 ±0.02 | 16.7 ±0.02 | 17.1 ±0.02 | 1.7 | 0.52 |
| A-FRCNN-16 (sum) | **16.2** ±0.02 | **16.7** ±0.01 | **14.0** ±0.01 | **14.6** ±0.02 | **17.9** ±0.01 | **18.3** ±0.01 | 1.7 | 0.98 |

significantly more computational expense. In addition, Sepformer had significantly more parameters than A-FRCNN-16.

**Efficiency.** We have seen that the three models based on intra- and inter-chunk framework, DualPathRNN, Gated DualPathRNN and Sepformer obtained SOTA SI-SNRi and SDRi values on WHAM! and WSJ0-2Mix datasets. We then investigated the computing efficiency of these models.

First, from Table 3 we see that DualPathRNN was 3× slower than A-FRCNN-16 during inference. Second, training DualPathRNN was also very slow. Training DualPathRNN on Libri2Mix took 142 hours on 8 GPUs while training A-FRCNN-16 took 39 hours (see Supplementary Materials for the training time of typical models). Third, Gated DualPathRNN and Sepformer were even slower than DualPathRNN.

We list more computational complexity metrics of Gated DualPathRNN, Sepformer and A-FRCNN-16 in Table 4, including the forward and backward GPU memory/time and GFlops of the forward pass by processing a four-second audio. We see that Gated DualPathRNN took about 3× and 2× more GPU memory than A-FRCNN-16 for forward and backward passes, respectively; Gated DualPathRNN took about 7× and 3× more GPU time than A-FRCNN-16 for forward and backward passes, respectively. Sepformer also took significantly more GPU memory and GPU time than A-FRCNN-16 during training. In addition, we found that the GFlops of A-FRCNN-16 could be significantly reduced when the concatenation was replaced with summation for information fusion, while SI-SNRi value dropped only slightly. In fact, A-FRCNN-16 (sum) took about 6× less GFlops than Gated DualPathRNN and Sepformer. In terms of these efficiency metrics, SuDoRM-RF 2.5x performed as well as A-FRCNN-16 (sum).

Taken together, compared with other SOTA models, A-FRCNN achieved good balance between speech separation accuracy and computational efficiency, and it can obtain good separation results with limited computing resource.

Table 4: Computational complexity metrics of different SOTA models.

| Model | F/B GPU Memory (GB) | F/B GPU Time (ms) | GFlops |
|---|---|---|---|
| Gated DualPathRNN | 0.43/7.78 | 503.8/893.2 | 125.28 |
| Sepformer | 0.62/11.3 | 311.8/702.1 | 145.58 |
| SuDoRM-RF 2.5x | 0.10/4.85 | 77.2/210.3 | 19.8 |
| A-FRCNN-16 | 0.14/4.07 | 74.4/263.5 | 123.3 |
| A-FRCNN-16 (sum) | 0.13/3.54 | 70.2/223.3 | 22.8 |

## 5.3 Ablation Study

The experiments were on the Libri2Mix dataset. We studied the influence of the number of stages $S$ (Figure 1c) by fixing the unfolding method to SC. The results was the best with $S = 5$ (Table 5). We then compared the results with different unfolding methods (Figure 4) by fixing $S = 5$ and found that the SC method was the best (Table 6). In the two experiments the A-FRCNN was unfolded for 8 time steps. We therefore used $S = 5$ and the SC method in all other experiments. Only the results with this setting are average results over 5 different runs in Tables 5 and 6; and we did not train models with other settings for multiple times considering the small standard deviations in previous tables.

Table 5: Test results of the A-FRCNN with different number of stages $S$.

| $S$ | SI-SNRi | SDRi | Params |
|---|---|---|---|
| 3 | 13.8 | 14.5 | 4.0M |
| 4 | 14.4 | 14.9 | 5.1M |
| 5 | **15.5** | **15.9** | 6.1M |
| 6 | 14.4 | 14.9 | 7.2M |

Table 6: Test results of A-FRCNN with different unfolding methods in the macro-level.

| Method | SI-SNRi | SDRi | Params | Time (s) |
|---|---|---|---|---|
| DC | 12.8 | 11.4 | 6.1M | 0.70 |
| CC | 14.3 | 14.8 | 6.7M | 0.96 |
| SC | **15.5** | **15.9** | 6.1M | 0.72 |

## 5.4 Transfer to DualPathRNN and Sandglasset

Both DualPathRNN [20] and Sandglasset [13] use intra-chunk and inter-chunk operations repeatedly to fuse information. We used our proposed MSF method in A-FRCNN to fuse 5 multi-scale intra-chunks and sent the result to the inter-chunk in one block, and repeated multiple blocks with tied weights. The new models, called A-DualPathRNN and A-Sandglasset, obtained better results than the original models, further demonstrating the merit of the proposed MSF method. Details can be found in Supplementary Materials.

## 6 Conclusion

We investigated an asynchronous updating scheme for a bio-inspired FRCNN architecture for solving the speech separation task. The resulted model A-FRCNN achieved better results than the conventional synchronous updating scheme in experiments. Compared with the models based on DualPathRNN, the proposed A-FRCNN achieved inferior results but was significantly more efficient in both training and inference, therefore achieved a good trade-off between accuracy and efficiency.

This work has some limitations. First, for fair comparison with many existing models, we did not consider reverberation, which we have to deal with in the real world. It is unknown how reverberation will influence the performance of our proposed model. Second, we only considered two speakers in the mixed speech and did not consider the more natural setting where the number of speakers is unknown. Third, the proposed unfolding scheme was obtained by hand. It would be interesting to use data-driven learning such as the neural architecture search techniques to determine the best design.

## Acknowledgments and Disclosure of Funding

We thank the authors of the Sandglasset model (Max W. Y. Lam and Jun Wang) for providing us the model code and some constructive comments about training strategies. This work was supported in part by the National Key Research and Development Program of China (No. 2017YFA0700904),

the National Natural Science Foundation of China (Nos. 62061136001, 61836014, U19B2034 and 61620106010) and the Tsinghua-Toyota Joint Research Fund.

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
