# OpenReview forum: "Speech Separation Using an Asynchronous Fully Recurrent Convolutional Neural Network"
_NeurIPS.cc/2021/Conference — NeurIPS 2021 Poster_

### Official Review · Reviewer_EiDq · 2021-06-27

**Rating:** 3
**Confidence:** 5

**Summary:**

The paper introduce a novel method to separate mixture of  sounds. The paper introduce a new neural architecture called Fully Recurrent Convolutional Neural Network (FRCNN) which use neural network with lateral connections. The main variant of the proposed method is the A-FRCNN which use asynchronous updating. The main novel parts are: (1) novel bottleneck structure (2) removing some connection that seems to be redundant.

**Limitations And Societal Impact:**

The authors discuss the limitation of the proposed method.

**Main Review:**

The paper has interesting ideas however if found that it has marginal contribution to the community due to the following:

1. The proposed architecture is very similar to the previous one in the literature ״Bridging the Gaps Between Residual Learning, Recurrent Neural Networks and Visual Cortex״ [https://arxiv.org/pdf/1604.03640.pdf]

2. The improvement is marginal for WHAM! dataset - less then 1dB. For Libri2Mix the results in the paper is weird: in all of previous papers the DPRNN method get better results then Conv-TasNet. Here, the author provide 14.1dB for DPRNN and 14.7dB for Conv-TasNet. I'm sure that with the right training procedure the DPRNN will get better results. Therefore, i don't think that the proposed method improve the results by more then 2dB.

3. The comparison is updated for 2 years ago, the DPRNN paper published at 2019. Furthermore, the paper luck of comparison with state-of-the-art results from previous publications such as Wavsplit [https://arxiv.org/pdf/2002.08933.pdf], Gated DPRNN [https://arxiv.org/pdf/2003.01531.pdf], SepFormer [https://arxiv.org/pdf/2010.13154v2.pdf], Sandglasset [https://arxiv.org/pdf/2010.13154v2.pdf]




**Time Spent Reviewing:**

4 hours

---

> ### Author Response · Authors · 2021-08-10
> **To Reviewer EiDq**
>
> **Q1: The proposed architecture is very similar to the previous one in the literature ״Bridging the Gaps Between Residual Learning, Recurrent Neural Networks and Visual Cortex״**
>
> A1: Thanks for raising up this point. The model in the suggested paper is our baseline, which is called S-FRCNN in the manuscript (also in Figure 1b). Our main contribution here is the change of the synchronous updating scheme to an asynchronous one (Lines 46-60). In addition, we have clarified the novelty of our proposed method by comparing it with S-FRCNN in the Response to IJCAI 2021 reviews. Please refer to the first point in the top of this webpage.
>
> **Q2: The improvement is marginal for WHAM! dataset - less then 1dB. For Libri2Mix the results in the paper is weird.**
>
> A2: On WHAM! Dataset, our A-FRCNN-16 obtained 14.53 dB SI-SNRi that is 0.8 dB higher than DPRNN. This improvement is not so large but significant considering that DPRNN obtained only 0.6 dB improvement on MSGT-TasNet (Dense) and nearly the same SI-SNRi as SudoRM-RF 2.5x. In addition, On Libri2Mix, A-FRCNN-16 obtained 2.55 dB higher SI-SNRi value on DPRNN. See Table 3 for details.
>
> We have stated in the paper that Conv-TasNet got 14.7 on LibriMix because it was trained on a larger dataset {train-360}, while all other models including DPRNN were trained on a smaller dataset {train-100}. The results of Conv-TasNet trained on {train-100} was 12.2, much lower than that of DPRNN 14.1.
>
> **Q3: The comparison is updated for 2 years ago. Furthermore, the paper luck of comparison with state-of-the-art results from previous publications.**
>
> A3: We agree that a comparison with more recent models is necessary for a comprehensive analysis on the performance for our proposed method. Given that both Wavesplit and Gated DPRNN apply speaker information as auxiliary training targets to achieve the best reported performance, here we selected Sandglasset as a representative of the more recent models and applied the proposed asynchronous update scheme to it (see our response A2 to Reviewer SsVq for details). How to further apply the proposed scheme to other recent architectures is left for future work.

---

> > ### Comment · Reviewer_EiDq · 2021-08-25
> > **Discussion**
> >
> > Thanks for the detailed response.
> >
> > I have a major concern regarding the results obtained in the proposed method. Libri2Mix is not a well-used dataset in the previous speech separation papers. Furthermore, for the WHAM dataset, the paper did not include Wavsplit results which are 15.4dB compared to the proposed method 14.53dB.
> >
> > Moreover, in the original submission, the paper did not include results for WSJ-2/3mix datasets due to computational constraints. WSJ-2mix dataset is similar to Libri2Mix/WHAM dataset in terms of training computational power. I don't see any reason why not to include results for a well-known dataset like this.
> >
> > I cannot advise to accept the since I'm not sure it obtain state-of-the-art results for speech separation. Unfortunately, I need to lower the score to "3: A clear reject."

---

> > > ### Author Response · Authors · 2021-08-25
> > > **Reply**
> > >
> > > Maybe we didn't make things clear.
> > >
> > > (1) We present results on the WHAM! dataset in the main text, which is more natural and challenging than WSJ0-2mix as it is constructed by adding noise to WSJ0-2mix. In addition, we present the results on WSJ0-2mix in the Supplementary Materials (Sec 1). It is seen that our proposed method achieved slightly inferior results than DualPathRNN on this dataset, but it runs 3 times faster than DualPathRNN. Due to the space limit we put it in the Supplementary Materials.
> > >
> > > (2) As we said in the previous response, Wavesplit solves speech separation in a different setting: it requires to use the speaker identity information for training. All methods compared in our paper do not use this information. Direct comparison between those methods and Wavesplit is unfair as our main focus is the model architecture, not which additional information should be exploited to improve the performance on this particular task. But we do think Wavesplit is a good method and will add its results in our paper (by indicating that it uses additional information).
> > >
> > > We beg you to read the paper (including Supplementary Materials) and our response again and reconsider your recommendation. Thanks a lot for your valuable time.

---

> > > > ### Comment · Reviewer_EiDq · 2021-08-25
> > > > **Discussion**
> > > >
> > > > Thank you for the quick replay.
> > > >
> > > > I just read the supplementary file it just fit my decision:
> > > > 1. WSJ-2mix: the proposed method achieve 18.26dB SISNR comparing to: Sepformer: 20.4dB, VSUNOS[Voice Separation with an Unknown Number of Multiple Speakers]: 20.1dB.
> > > > 2. WHAM: the proposed method achieve 14.53dB SISNR comparing to: VSUNOS: 15.2dB and i'm sure that Sepformer get also better results.
> > > >
> > > > This make me wonder what is the gap between the proposed method results and the Sepfoermer/VSUNOS for the Libri2mix dataset.
> > > >
> > > > Both method don't use speaker identity during the training.
> > > > I'm sorry for this but i cannot recommend a paper that the results is lower then papers that publish over 1+ year ago at top tier conferences.

---

> > > > > ### Author Response · Authors · 2021-08-28
> > > > > **Our model is significantly more efficient than VSUNOS and Sepformer**
> > > > >
> > > > > There are two points that we would like to clarify.
> > > > >
> > > > > (1) Although we do agree that the SI-SNRi numbers are important indicators of the capacity of the models, we would also like to highlight the importance of the tradeoff between the model size, model complexity (in terms of training/inference speed and memory footprints), and model performance. For the models mentioned in your latest reply:
> > > > >
> > > > > a) Sepformer requires a training batch size of 1 on a Nvidia V100 GPU to obtain the reported performance (Paragraph 4, Section 3.2 in the Sepformer paper, also mentioned in A4 in the response to Reviewer sPgG). The number of parameters is 26M (Table 1, Section 4.1 in the Sepformer paper). We have mentioned in the response to Reviewer sPgP about this issue on computational resource required for training (see A4).
> > > > >
> > > > > b) VSUNOS (or Gated DPRNN in your first review comments) proposed the gated DPRNN architecture and thus had a higher model complexity than the standard DPRNN. We have mentioned in the response to Reviewer sPgG about the training time required for standard DPRNN model and the proposed A-FRCNN models (see Limitations and Societal Impact, 3x faster).
> > > > >
> > > > > As advised by Reviewer sPgG, we report the efficiency and performance of the newer SOTA models (Sepformer and VSUNOS) and our proposed model on WSJ0-2Mix in the table below. We can see that Sepformer requires ~5-6x more GPU memory, ~2x more GPU time, and ~3x more CPU time than A-FRCNN for inference; while VSUNOS requires ~3-4x more GPU memory, ~2x more GPU time, and ~5x more CPU time than A-FRCNN for inference. In addition, when summation instead of concatenation is used to fuse different branches, our proposed model (last row) requires about 6-7x less GFlops than Sepformer and VSUNOS.
> > > > >
> > > > > | Model | Forward/Backward GPU memory (GB) | Forward/Backward GPU Time (ms) | CPU Inference Time (s) | Params (M) | GFlops | SI-SDRi | SDRi |
> > > > > | --- | --- | --- | --- | --- | --- | --- | --- |
> > > > > | Sepformer | 0.20/2.90 | 128.5/343.7 | 1.02 | 26 | 42.74 | 20.4 | 20.5 |
> > > > > | VSUNOS | 0.13/2.79 | 141.5/397.9 | 1.46 | 7.5 | 32.30 | 20.1 | - |
> > > > > | SudoRM-RF 1.0x | 0.03/0.55 | 40.7/186.0 | 0.24 | 2.7 | 9.15 | 17.1 | 17.3 |
> > > > > | AFRCNN-16 | 0.04/1.07 | 72.0/223.4 | 0.30 | 6.1 | 35.08 | 18.3 | 18.6 |
> > > > > | AFRCNN-16 (sum) | 0.03/0.88 | 64.4/185.4 | 0.28 | 1.69 | 5.71 | 17.9 | 18.3 |
> > > > >
> > > > > You asked “This make me wonder what is the gap between the proposed method results and the Sepformer/VSUNOS for the Libri2mix dataset”, and we would like to argue that the model complexity is an important factor and all performance comparisons should take it into account.
> > > > >
> > > > > Moreover, we would like to also mention that the SudoRM-RF model, which was published in mid 2020 and cited the DPRNN and Wavesplit model as reference [3] and [25] in its paper, respectively, did *not* outperform the best reported numbers of these two models, but had a significantly lower computational requirement (Table 1, Section 4 in the SudoRM-RF paper). We have mentioned in the previous responses that the proposed A-FRCNN has similar network designs on the downsampling and upsampling blocks as SudoRM-RF (A0 & A1, response to Reviewer sPgG), and we can obtain higher separation performance than the SudoRM-RF baselines with an even fewer computational requirement (Table 2, A3, response to Reviewer sPgG). We thus argue that achieving SOTA SI-SNRi numbers or not should not be the *only* criteria for network architectures that are significantly more efficient and lightweight.
> > > > >
> > > > > (2) You have mentioned that VSUNOS did not use speaker identity during training. Unfortunately this is not accurate, as Figure 3 and Eqn. (6) in the VSUNOS paper clearly showed that the speaker ID loss was applied during training, and Table 2 in the VSUNOS paper showed that the ID loss was important for the model to achieve its best reported performance. We have already explained the reasons why we did not select models that use auxiliary speaker-information-related training objectives in our previous responses.

---

> > > > > > ### Comment · Reviewer_EiDq · 2021-08-30
> > > > > > **Discussion**
> > > > > >
> > > > > > (2) Please read the paper VSUNOS carefully: without the ID loss (speaker identity), VSUNOS gets 19.76dB [wsj2mix] which is still better then the proposed method - 18.26dB.

---

> > > > > > > ### Author Response · Authors · 2021-08-30
> > > > > > > **reply**
> > > > > > >
> > > > > > > We agree that VSUNOS achieved better performance. As we said, when evaluating a model, one should balance the performance and efficiency, because different models have different application scenarios.  VSUNOS requires ~3-4x more GPU memory, ~2x more GPU time, and ~5x more CPU time than A-FRCNN for inference. Please see (1) for details. We think you are aware of this point, but don't understand why you are so fascinated with SOTA performance and ignore the efficiency.  We have this impression because most of your replies are about SOTA, and you even lower the rating to REJECT because we didn't achieve the SOTA performance.
> > > > > > >
> > > > > > > In addition, please also consider the novelty and potential influence of the proposed method. To the best of our knowledge, we for the first time show that by using asynchronous updating scheme, an RNN can achieve better performance than the conventional synchronous updating scheme.
> > > > > > >
> > > > > > > No matter you will change your rating or not, we would like to thank you anyway for spending so much time on reading our manuscript and discussing with us.

---

### Official Review · Reviewer_sPgG · 2021-07-15

**Rating:** 7
**Confidence:** 5

**Summary:**

The paper proposes a new separation module for extracting multi-scale features and aggregating them for improving the performance of source separation models. The proposed architecture consists of an asynchronous update mechanism which operates at different time-scales and fuses the features in parallel at the latter stage. The authors claim that the inspiration behind their proposed model is the non-simultaneous stimulus of human neurons when they get triggered from some sensory activity. The authors provide quantitative results on widely used benchmarks datasets and show improvement over strong baselines. Finally, the authors conduct ablations studies to show the importance of each update scheme inside the proposed separation module block.

**Limitations And Societal Impact:**

The authors do vaguely discuss a few limitations of their work in the conclusion section but they can also include any of the points that I raised here (if they do not wish to address them in the rebuttal).

In my opinion, this work raises some potential negative societal impacts, especially environmental ones, that the authors do not discuss. For instance, the hours required to train all those models could be at least be stated in the appendix and how the authors improve upon the training time of the baseline models.

**Main Review:**

### Pros
This is a well written paper, the authors have put significant effort into presenting in detail their proposed architecture both in text and with explanatory figures showing the differences between different update mechanisms. Moreover, the parallel with the biological aspect of how the neurons work is quite interesting and is something that is missing from many current models in the audio source separation community. The experiments seems quite extensive and solid by using multiple baselines and evaluating different variations of the proposed architecture. This is definitely a quality paper in terms of the experimental framework, the evaluation and the derivations. Overall, I believe that the paper has great potential if some concerns are addressed and could contribute significantly to the field of source separation.

### Limitations / Criticism
However, the paper has several limitations which downplay the true potential of the paper. Please see below my concerns and questions to the authors in descending order of importance (0 is the most important):

0. The authors claim that the inspiration of the proposed architecture comes from the morphology of the human neural activations. However, the submitted code is weirdly similar to the original implementation of the SuDo RM-RF [24] model which seems to be the backbone structure of the proposed models (in fact after checking with an online tool, the code for control 1 and 2 has > 80% **identical** lines with this file https://github.com/etzinis/sudo_rm_rf/blob/master/sudo_rm_rf/dnn/models/improved_sudormrf.py but also the proposed A-FRCNN has the same level of similarity). Although, using open source code is encouraged (**if it’s accredited appropriately**), in this specific case, it downplays the inspiration from the biological aspect and one could misinterpret the proposed architecture as an ad-hoc variation of a SuDo RM-RF [24] model.
1. The authors could build the inspiration of their module architecture by first identifying the limitations of the SuDo-RM-RF [24] and then identifying biologically plausible solutions. Moreover, I believe that such story would make more sense and it would definitely avoid the negative affect on the originality of the paper, especially to a reader who is knowledgeable wrt the SuDo-RM-RF [24] architecture and can immediately see the similarities of the two works. Also, not that important, but why did the authors prefer to use the Asteroid [18] implementations for training all the models when they had access to the original repository of SuDo-RM-RF [24] and other models?
2. The authors need to delve deeper into the details of their biologically inspire model. Specifically, I can find only two references wrt the biological inspiration [1, 3], whereas I can see whole paragraphs (Lines 34-45) where multiple things stated should be backed up by the corresponding citations. By doing so, and in combination with the limitations of used separation models, the authors can pass a clear message to the reader on why such biological mechanisms should be employed towards building better architectures.
3. Since the authors refer to the efficiency of their model (e.g. Line 13) and they also use the term “light” for their proposed architecture, I have the following suggestions. It is quite convenient to put one number (e.g. the number of parameters or the inference time) and compare the efficiency. However, the authors propose a complex architecture that might severely enlarge the actual memory footprint of a forward pass of the network (or backward) mainly because of the intermediate activations tensors. Moreover, the same could also apply for the FLOPs or any other computational aspect. The authors should also report several other factors like actual memory requirements, time consuming on training on GPU / CPU and floating point operations and compare with the SuDo-RM-RF [24]. Based on that, the group communication mechanism [1*] has been shown to be able to significantly reduce the number of trainable parameters by dividing the channels into groups and processing them independently (with shared parameters across the groups). The authors could also integrate it to their proposed method and compare their numbers since it has also been shown beneficial for the SuDo-RM-RF [24] and DPRNN [14] models in [1*, 2*].
4. The authors should also compare against stronger baselines with respect to the upper bound of the separation performance. A stronger SOTA architecture could be the transformer architecture proposed in [3*], which also has an open source repository.
5. Lines 221-222: “Scale invariance is ensured by normalizing and zero-mean before the calculation.” There is no need to normalize the targets and the estimates since scale-invariant signal-to-noise ratio (SI-SNR) [4*] is by itself scale-invariant. The weirdly noted term  $ A_{\operatorname{target}} $ in equation 7 ensures that the magnitude of the projection of the estimated source $ \hat{s} $ onto the target $ s $ would be irrelevant to SI-SDR (since $ A_{\operatorname{target}} $  is a vector it should probably be lowercase).
6. The citation for the SI-SDR in Line 217 is incorrect, it should be [4*] instead of [26] which is the SDR paper.
7. Differences smaller than <0.1 dB are not either significant nor hearable, thus I would suggest to round up all those performance numbers to a one decimal precision (it would also make the Tables less cluttered).
8. Lines 82-83. "For instance, Sudo-RM-RF [24] uses a U-Net structure [20], obtaining good results on smaller-sized models". This is not true, SuDo-RM-RF uses a repetitive U-ConvBlock structure. Now, each U-ConvBlock uses a U-Net architecture. "Smaller sized models" is also loosely defined considering the efficiency analysis I also referred above (See point 3.) where the number of parameters is not the major concern.

Typos that I found:
Line <>:
159: Therefore → Therefore,
164: increase → increases
264: wham! → WHAM!

I am willing to increase my score if most of the most important above concerns are addressed by the authors since I truly believe that the paper has a great potential to change the way how we design neural network architectures for source separation.

[1] Mark Bear, Barry Connors, and Michael A Paradiso. Neuroscience: Exploring the Brain. Jones & Bartlett Learning, LLC, 2020.

[3] Peter Dayan and Laurence F Abbott. Theoretical Neuroscience: Computational and Mathemati362 cal Modeling of Neural Systems. The MIT Press, 2001.

[14] Yi Luo, Zhuo Chen, and Takuya Yoshioka. Dual-path rnn: efficient long sequence modeling for time-domain single-channel speech separation. In Proceedings of the International Conference on Acoustics, Speech and Signal Processing (ICASSP), pages 46–50, 2020.

[18] Manuel Pariente, et. al.. Asteroid: the PyTorch-based audio source separation toolkit for researchers. In Proceedings of the International Speech Communication Association (Interspeech), 2020.

[24] Efthymios Tzinis, Zhepei Wang, and Paris Smaragdis. Sudo rm-rf: Efficient networks for universal audio source separation. In Proceedings of the Machine Learning for Signal Processing (MLSP), pages 1–6, 2020.

[26] Emmanuel Vincent, Rémi Gribonval, and Cédric Fevotte. Performance measurement in blind audio source separation. IEEE/ACM Transactions on Audio, Speech, and Language Processing, 14(4):1462–1469, 2006.

[1*] Luo Y, Han C, Mesgarani N. Ultra-lightweight speech separation via group communication. InICASSP 2021-2021 IEEE International Conference on Acoustics, Speech and Signal Processing (ICASSP) 2021 Jun 6 (pp. 16-20). IEEE.

[2*] Tzinis E, Wang Z, Jiang X, Smaragdis P. Compute and memory efficient universal sound source separation. arXiv preprint arXiv:2103.02644. 2021.

[3*] Subakan C, Ravanelli M, Cornell S, Bronzi M, Zhong J. Attention is all you need in speech separation. InICASSP 2021-2021 IEEE International Conference on Acoustics, Speech and Signal Processing (ICASSP) 2021 Jun 6 (pp. 21-25). IEEE.

[4*] Le Roux J, Wisdom S, Erdogan H, Hershey JR. SDR–half-baked or well done?. InICASSP 2019-2019 IEEE International Conference on Acoustics, Speech and Signal Processing (ICASSP) 2019 May 12 (pp. 626-630). IEEE.

**Time Spent Reviewing:**

5 hours

---

> ### Author Response · Authors · 2021-08-10
> **To Reviewer sPgG**
>
> **Q0: The code for A-FRCNN is similar to SudoRM-RF.**
>
> A0: Thanks for pointing this out and we are very sorry about our negligence. To start the project, we needed to find a strong baseline model that is based on CNN and we found that SudoRM-RF suited our needs. Then we adopted the existing codebase, modify the connectivity patterns and update schemes upon it, but only added reference to the SudoRM-RF paper ([24]) instead of the codebase. We sincerely apologize for not assigning proper credits to it and will add that in our revision.
>
> **Q1: About the organization of the paper and Asteroid.**
>
> A1: After discussion, we agree that a reorganization is necessary in the revision. We plan to first discuss different multiscale methods including SudoRM-RF and identify their limitations (mainly in the lack of efficient information interaction between different scales), and then provide a novel view of them by treating them as special cases of the FRCNN framework with different asynchronous updating schemes. Then we raise the main argument of the paper, which is to find more efficient and effective “asynchronous” updating schemes in such network architectures. Further suggestions about the organization are welcome.
>
> We chose Asteroid as the platform to implement our models mainly for its easiness to use and its coverage on various existing models, as we compared our methods to multiple other systems in our manuscript. As for the SudoRM-RF model, we found that the implementation in Asteroid is almost identical to the one in the codebase released by the authors. On the WSJ0-2Mix dataset, the SI-SNRi values of different versions of SudoRM-RF obtained by using Asteroid were nearly the same as those reported in the original paper [24] (see also our Table S1)—the difference was within 0.1 dB. In addition, we found that the implementation of some methods often yielded better results than reported in the original papers. E.g., on the WSJ0-2Mix dataset, the SI-SNRi values:
>
> Conv-TasNet: 16.2 (Asteroid) VS 15.31 (original paper [16])
>
> DPRNN: 19.3 (Asteroid) VS 18.8 (original paper [14]).
>
>
> **Q2: Delve deeper into the details of their biologically inspire model.**
>
> A2: Thanks for raising this important discussion. There are two major messages in Lines 34-45. First, “the neurons along a sensory hierarchy do not ﬁre simultaneously like shown in Figure 1b.” We will add the following arguments to support this claim: For example, it was reported that the neural response initialized at a retinotopic position in anesthetized rat V1 propagated uniformly in all directions with a velocity of 50–70 mm/s, slowed down at the V1/V2 area border, after a short interval, spread in V2, then reflected back in V1 [1&]. In addition, it is estimated that the traveling speed of the neural signal in the brain ranges from 2 miles per hour to 200 or more miles per hour and the difference is mainly due to the type of fiber [2&].
>
> Second, “the history of ANN development indicates that getting inspiration from the brain is enough to make great progress if task-speciﬁc techniques are combined.” We will add the following arguments to support this claim: Inspired by the discovery of simple cells and complex cells in cat visual cortex [3&][4&], a hierarchical model Neocognitron [5&] was proposed and later developed into convolutional neural networks [6&] by applying the backpropagation algorithm.
>
> [1&] Xu, W., Huang, X., Takagaki, K., and Wu, J.-y. (2007). Compression and reflection of visually evoked cortical waves. Neuron, 55, 119–129.
>
> [2&] Myers, David G. Psychology, 4th Edition. New York: Worth Publishers Inc,1995: 43.
>
> [3&] Hubel, D. H. (1959). Single unit activity in striate cortex of unrestrained cats. Journal of Physiology-London, 147(2), 226-238.
>
> [4&] Hubel, D. H., & Wiesel, T. N. (1962). Receptive fields, binocular interaction and functional architecture in cats visual cortex. Journal of Physiology-London, 160(1), 106-154.
>
> [5&] Fukushima, K. (1980). Neocognitron - a self-organizing neural network model for a mechanism of pattern-recognition unaffected by shift in position. Biological Cybernetics, 36(4), 193-202.
>
> [6&] LeCun, Y., Boser, B., Denker, J. S., Henderson, D., Howard, R. E., Hubbard, W., & Jackel, L. D. (1989). Backpropagation applied to handwritten zip code recognition. Neural Computation, 1(4), 541-551.
>
> **Q3: Test actual memory requirements, GPU / CPU and floating point training time. Use the group communication mechanism and integrate it into A-FRCNN and compare the results.**
>
> A3: We tested the models on the same computer server by inputting a 1-second speech segment from the WSJ0-2Mix dataset and obtained the numbers as follows (SI-SNRi values are copied from Table S1)
>
> | Model | GFlops | Forward/Backward GPU memory (GB) | SI-SNRi (dB)|
> | --- | --- | --- | --- |
> | SudoRM-RF 0.25x| 1.04 | 0.30/0.25 | 13.4 |
> | SudoRM-RF 0.5x  | 1.51 | 0.40/0.45 | 15.4 |
> | SudoRM-RF 1.0x| 2.54 | 0.61/0.86 | 17.1 |
> | A-FRCNN-4         | 9.15 | 0.30/0.30 | 15.6 |
> | A-FRCNN-8         | 17.79 | 0.56/0.56 | 17.1 |
> | A-FRCNN-16       |35.08 | 1.07/1.07 | 18.3 |
>
> It is seen that A-FRCNN-8 obtained the same SI-SNRi value as SudoRM-RF 1.0x with a smaller memory footprint but higher number of FLOPs. We found that the concatenation operation for integrating the information from different scales (Figure 1d) contributed to most of the overall model complexity. We thus modified the “concatenation” operation to the “summation” operation, given that the feature dimensions at different scales are set to the same across the entire pipeline. Compared to the original design, this modification gave us up to 12.7× FLOPs decrease and 1.5×memory consumption decrease while only resulted in a minor performance degradation:
>
> | Model | GFlops | Forward/Backward GPU memory (GB) | SI-SNRi (dB)|
> | --- | --- | --- | --- |
> | SudoRM-RF 0.5x  | 1.51 | 0.40/0.45 | 15.4 |
> | A-FRCNN-8      | 17.79 | 0.56/0.56 | 17.1 |
> | A-FRCNN-8 (sum)  |1.41 | 0.36/0.36 | 16.7 |
>
> While the original A-FRCNN has a significantly higher FLOPs and slightly higher GPU memory consumption than the SudoRM-RF 0.5x model, using the “summation” operation for multiscale feature fusion enables the model to have even lower FLOPs and GPU memory consumptions than the SudoRM-RF 0.5x with only a 0.4 dB performance degradation. This modification further proves the advantage of our proposed design. However, due to the very limited time in the rebuttal period, we did not manage to update other experiment results under this modification, but we will continue running the experiments and add the results to all tables.
>
> We would like to clarify that the group communication (GC) method was officially published after our initial submission, hence we did not apply this to our method. To validate its performance, here we apply GC to both SudoRM-RF and A-FRCNN models with a group size of 16:
>
> | Model | Params | GFlops | Forward/Backward GPU memory (GB) | SI-SNRi (dB)|
> | --- | --- | --- | --- | --- |
> | SudoRM-RF 0.5x-GC  | 303.57K | 0.83 | 0.61/0.61 | 12.5 |
> | A-FRCNN-8-GC      | 286.24K | 1.95 | 0.67/0.67 | 12.7 |
>
> We can observe a significant performance degradation compared to the plain models. The GC paper stated that a deeper model is necessary to maintain the performance when a large group size is selected. Finding the best hyperparameter configuration for GC-equipped models is left for future works.
>
> **Q4: Compare against stronger baselines with respect to the upper bound of the separation performance.**
> A4: The main contribution of our paper is the proposed connection scheme (Figure 1c) which is not limited to specific model structure. Theoretically, one can replace CNNs or RNNs (Section 2 in Supplementary Materials) in our method with transformers and treat them as nodes in Figure 1c. A main issue for transformer-based SOTA models is that they typically require tremendous computational resources for training. For example, the SepFormer model in [3*] stated that a batch size of 1 was used and “each epoch takes approximately 1.5 hours on a single NVIDIA V100 GPU with 32 GB of memory”. Given the limited computational resources, we selected another transformer-based system, the Sandglasset (Lam, Wang, Su, et al. ICASSP 2021), for the evaluation of the proposed scheme on SOTA systems. The resulting model, A-Sandglasset, obtained 0.6 dB higher SI-SNRi than the plain Sandglasset on the WSJ0-2Mix dataset. See our response A2 to Reviewer SsVq for details.
>
> **Q5: “Scale invariance is ensured by normalizing and zero-mean before the calculation.” This is weird.**
>
> A5: We are sorry for this inaccurate statement and will remove this in our revision. We would like to clarify that the calculation of SI-SNR in all our experiments is correct and all the experiment results are valid.
>
>
> **Q6: The citation for the SI-SDR in Line 217 is incorrect.**
>
> A6:  Thanks for pointing it out and we will fix it in our revision.
>
> **Q7: Round up all those performance numbers to a one decimal precision.**
>
> A7: Thanks for mentioning this and we agree that one decimal precision should be enough. We will modify all results in out revision.
>
> **Q8: Problems in “For instance, Sudo-RM-RF [24] uses a U-Net structure [20], obtaining good results on smaller-sized models.”**
>
> A8: We will change this sentence to: For instance, SudoRF-RF [24] uses repetitive U-Nets [20], obtaining good results with high efficiency.
>
> **Limitations and Societal Impact:**
> We will list the training time for all models in the Supplementary Materials. The training time for several typical models on Libri2Mix is listed below.
>
> | Model | Training time (h)|
> | --- | --- |
> | Conv-TasNet {train-360}| 50.34 |
> | Conv-TasNet {train-100}| 14.16 |
> | SudoRM-RF 0.25x     | 12.50 |
> | SudoRM-RF 0.5x      | 14.20 |
> | SudoRM-RF 1.0x      | 19.65 |
> | DPRNN             | 142.51 |
> | A-FRCNN-4          | 14.77 |
> | A-FRCNN-8          | 22.79 |
> | A-FRCNN-16         | 39.44 |

---

> > ### Comment · Reviewer_sPgG · 2021-08-21
> > **Reply**
> >
> > I would like to thank the authors for their extensive reply and the assurance that they will address my concerns in the next version. If all actions are taken for addressing my concerns, as ditto, I will increase my score.
> >
> > There is one mistake in your computations since the forward time cannot be the same with a backward pass (backward = forward + backprop). There is probably a small bug in the profiler code that can be easily fixed.
> >
> > For example, those numbers are wrong for the A-FRCNN models.
> >
> > Model	GFlops	Forward/Backward GPU memory (GB)	SI-SNRi (dB)
> >
> > SudoRM-RF 0.25x	1.04	0.30/0.25	13.4
> >
> > SudoRM-RF 0.5x	1.51	0.40/0.45	15.4
> >
> > SudoRM-RF 1.0x	2.54	0.61/0.86	17.1
> >
> > A-FRCNN-4	9.15	**0.30/0.30**	15.6
> >
> > A-FRCNN-8	17.79	**0.56/0.56**	17.1
> >
> > A-FRCNN-16	35.08	**1.07/1.07**	18.3

---

> > > ### Author Response · Authors · 2021-08-24
> > > **Reply**
> > >
> > > Definitely, we will revise the paper according to your suggestions. We welcome any further suggestions or comments.
> > >
> > > About the GPU memory reported in the response, thank you for pointing out this mistake. Please note that we reported GPU memory instead of GPU time in our previous response. The GPU memory numbers in our previous response are incorrect. They were obtained by using the SudoRM-RF open source code:
> > > https://github.com/etzinis/sudo_rm_rf/blob/master/sudo_rm_rf/utils/extract_model_performance.py.
> > > We found that the forward/backward GPU memory numbers were included in the output of the codes, so we reported these numbers in our response. Unfortunately, as reminded by you, we checked the code and found two problems in the code. (1) When getting the peak memory for the backward process, the code does not clear the peak memory used in the forward process. That’s why in our results the forward and backward memory numbers were the same. The correct method is to use torch.cuda.reset_peak_memory_stats() to reset the peak memory after getting the maximum GPU memory for the forward process. (2) The code doesn’t use torch.no_grad() in the forward calculation to remove the gradient information. This overestimates the GPU memory needed in the forward process (inference only). This might explain why the forward GPU memory for SudoRM is always large (sometimes even larger than the backward GPU memory) as reported in [24].
> > >
> > > We are sorry for our carelessness. We have modified the source code and updated the GPU memory numbers in all of the three tables in our previous response. See below. We also report GPU time now.
> > >
> > > | Model | Forward/Backward GPU memory (GB) | Forward/Backward GPU Time (ms) |
> > > | --- | --- | --- |
> > > | SudoRM-RF 0.25x| 0.02/0.16 | 9.8/45.5 |
> > > | SudoRM-RF 0.5x | 0.02/0.29 | 18.9/99.7 |
> > > | SudoRM-RF 1.0x| 0.3/0.55 | 40.7/186.0 |
> > > | A-FRCNN-4 | 0.04/0.34 | 18.9/70.0 |
> > > | A-FRCNN-8 | 0.04/0.58 | 32.9/127.7 |
> > > | A-FRCNN-16 | 0.04/1.07 | 72.0/223.4 |
> > >
> > > | Model | Forward/Backward GPU memory (GB) | Forward/Backward GPU Time (ms) |
> > > | --- | --- | --- |
> > > | SudoRM-RF 0.5x | 0.02/0.29 | 18.9/99.7 |
> > > | A-FRCNN-8 | 0.04/0.58 | 32.9/127.7 |
> > > | A-FRCNN-8(sum) |0.02/0.35 | 20.4/120.0 |
> > >
> > > | Model | Forward/Backward GPU memory (GB) | Forward/Backward GPU Time (ms) |
> > > | --- | --- | --- |
> > > | SudoRM-RF 0.5x-GC  | 0.02/0.32 | 30.5/114.9 |
> > > | A-FRCNN-8-GC      | 0.03/0.65 | 41.3/131.7 |

---

> > > > ### Comment · Reviewer_sPgG · 2021-08-25
> > > > **Reply 2**
> > > >
> > > > Thanks for figuring out the bug in the memory allocation for the forward pass. However, I do think that one has to include the max memory of the forward pass inside the backward GPU memory metric since the gradient graph allocation is required during training (thus, the backward memory utilization metrics were fine).  I am changing my evaluation to a weak accept rating.
> > > >
> > > > I would also suggest to include these metrics for the newer SOTA models (Sepformer and VSUNOS) alongside their SISDR performance in order to address EiDq's comments about the lower performance. I believe that a selling point for your paper could be the efficiency / separation performance trade off with respect to several metrics. AFAIK, 20dB vs 18.5dB of mean separation performance is negligible compared to obtaining 4x speedup or requiring 4x less memory footprint. However, in order to show that your proposed models are compelling, you need to include tables with SISDR alongside all the computational aspects and/or visualizing Paretto frontiers. If the proposed models have better performance than SudoRM-RF [24], score on par with the new SOTA models and also provide a significant speedup or significantly lower memory footprint, then I can increase my score to a solid acceptance rating.

---

> > > > > ### Author Response · Authors · 2021-08-28
> > > > > **Significantly more efficient than the newer SOTA models**
> > > > >
> > > > > We are sorry for any confusion caused by our description. We did include the max memory of the forward pass in the backward pass metric - the computational graph allocation cost, forward pass cost, and backpropagation cost were all considered in the reported numbers. What we corrected was the way we calculated the forward pass cost by setting "torch.no_grad()" to make sure that we were accurately reflecting the cost during the pure inference stage. The calculation of the backward cost was kept the same as the provided pipeline in SudoRM-RF codebase:
> > > > > https://github.com/etzinis/sudo_rm_rf/blob/fbd905a7c2a418fb38131bed651db4379760d105/sudo_rm_rf/utils/extract_model_performance.py#L176
> > > > >
> > > > > Briefly, the code is like this:
> > > > > ```python
> > > > > torch.cuda.reset_peak_memory_stats()
> > > > > def forward():
> > > > >     torch.no_grad()
> > > > >     model()
> > > > > torch.cuda.reset_peak_memory_stats()
> > > > > def backward():
> > > > >      model()
> > > > >      cal_loss()
> > > > >      backward()
> > > > > ```
> > > > >
> > > > > As indicated in the end of ABSTRACT, high efficiency is actually one of our selling points. Thank you very much for the suggestion about adding the efficiency metrics for newer SOTA models. The metrics for Sepformer, VSUNOS, SudoRM-RF and AFRCNN are summarized in the table below. We can see that Sepformer requires ~5-6x more GPU memory, ~2x more GPU time, and ~3x more CPU time than A-FRCNN for inference; while VSUNOS requires ~3-4x more GPU memory, ~2x more GPU time, and ~5x more CPU time than A-FRCNN for inference. In addition, when summation instead of concatenation is used to fuse different branches, our proposed model (last row) requires about 6-7x less GFlops than Sepformer and VSUNOS. Your guess about the significant higher efficiency of A-FRCNN compared with newer SOTA models is verified. In terms of these efficiency metrics, SudoRM-RF performed as well as AFRCNN.
> > > > >
> > > > > | Model | Forward/Backward GPU memory (GB) | Forward/Backward GPU Time (ms) | CPU Inference Time (s) | Params (M) | GFlops | SI-SDRi (dB) | SDRi (dB) |
> > > > > | --- | --- | --- | --- | --- | --- | --- | --- |
> > > > > | Sepformer | 0.20/2.90 | 128.5/343.7 | 1.02 | 26 | 42.74 | 20.4 | 20.5 |
> > > > > | VSUNOS | 0.13/2.79 | 141.5/397.9 | 1.46 | 7.5 | 32.30 | 20.1 | - |
> > > > > | SudoRM-RF 1.0x | 0.03/0.55 | 40.7/186.0 | 0.24 | 2.7 | 9.15 | 17.1 | 17.3 |
> > > > > | AFRCNN-16 | 0.04/1.07 | 72.0/223.4 | 0.30 | 6.1 | 35.08 | 18.3 | 18.6 |
> > > > > | AFRCNN-16 (sum) | 0.03/0.88 | 64.4/185.4 | 0.28 | 1.69 | 5.71 | 17.9 | 18.3 |
> > > > >
> > > > > BTW, VSUNOS achieved 20.1 dB SI-SDRi by using the speaker identity information which is not used by other models in the table.

---

> > > > > > ### Author Response · Authors · 2021-08-30
> > > > > > **To Reviewer #sPgG**
> > > > > >
> > > > > > To Reviewer #sPgG:
> > > > > >
> > > > > > Thanks for your valuable advice in the last reply. We have provided the efficiency metrics (e.g., memory footprint and forward/backward time) for A-FRCNN, SudoRM-RF, Sepformer and VSUNOS, and found that A-FRCNN and SudoRM-RF are significantly more efficient than the other two models. We welcome any further suggestions.
> > > > > >
> > > > > > We are not sure why Reviewer #EiDq even lowered the rating score from 5 to 3 (clearly rejection) after our reply, which is a huge drop. Then the average score of the paper now is around the borderline between acceptance and rejection.  Any help for improving the quality of the paper is welcome.

---

> > > > > > > ### Comment · Reviewer_sPgG · 2021-08-31
> > > > > > > **Reply 3**
> > > > > > >
> > > > > > > I can clearly see the effort of the authors during this rebuttal period as well as the improvement of the Paretto frontier (performance vs computrational aspects) over multiple important metrics. I updated my score since the proposed model seems to be improving upon the Sudo rm -rf models which is a strong baseline for obtaining good separation performance with limited computational resources and score on par with the latest SOTA models with uncostrained computational bottleneck.
> > > > > > >
> > > > > > > I also understand Reviewer's #EiDq concerns on the lower separation performance scores since the submitted version of the manuscript did not discuss the computational resource requirements (as it should) and focused mostly on one metric of the source separation. The authors should include the latest table posted in this thread in the main text and discuss how important is to consider all other metrics besides one mean performance SI-SDR score (I changed my score based on this premise).

---

> > > > > > > > ### Author Response · Authors · 2021-08-31
> > > > > > > > **Reply to Reviewer sPgG**
> > > > > > > >
> > > > > > > > Thanks a lot for your help. The latest tables posted in this thread will be added to the main text of the paper, together with discussion on the trade-off between separation performance, model size and computational efficiency. All other comments and suggestions from all reviewers will also be taken into account when we revise the paper.

---

### Official Review · Reviewer_UprN · 2021-07-16

**Rating:** 7
**Confidence:** 3

**Summary:**

The paper presents biologically inspired design of recurrent convolutional neural networks with bottom-up, lateral, and top-down connections. In contrast to a ‘synchronous’ update of all nodes according to connectivity, the biologically motivated unrolling of recurrent connections and information flow in time results in an ‘asynchronous’ update where different nodes get updated at different time steps with bottom-up, lateral, and top-down information flow.  These architectures are termed A-FRCNN (asynchronous fully recurrent convolutional neural networks).  A-FRCNN architectures can equivalently be viewed as feed-forward networks with specific weight tying.  These models are applied to the task of speech separation and state of the art results are obtained on two data sets.

**Ethical Concerns:**

No ethical concerns I can think of.

**Limitations And Societal Impact:**

Limitations adequately addressed, and no negative societal impact I can think of.

**Main Review:**

Overall, the paper is well written with clear descriptions and explanations and presents strong results.  A few clarifying questions / comments:

* At inference time I’m assuming (but couldn’t confirm in the paper) the A-FRCNN architecture is applied as well, or do the A-FRCNN architectures only guide how the FRCNN model is trained and then at inference time the fully connected FRCNN model is used in synchronous mode?  If it is the former, that is A-FRCNN architecture is used for both training and inference, then the model architecture (and not just parameter updates at training time) is different from FRCNN, in which case is it fair to call the proposed approach asynchronous updating scheme of FRCNN?

* Given the biological inspiration for A-FRCNN, a control (aside from the two controls presented in the paper) that seems missing is a model that is similar to S-FRCNN but is without all the skip connections (thus restricting information flow to strictly one layer in one time step).  Did the authors try this model, or any specific reason for not trying it?  Similarly could there be variants where skip connections (both bottom-up and top-down) will skip in the time dimension (a connection between node N and N+2 will result in an unfolded model that connects node N at time T to N+2 at time T+2)?

* For a given FRCNN network, the proposed approach of asynchronous updates and corresponding network design essentially adds another dimension of possibilities to explore.  Can there be a data driven learning based approach to determine optimal design, or at least guide towards that?  This seems important, because as the results with various asynchronous configurations show, the gains are quite sensitive to the model architecture (and hence it is quite possible that substantially better configurations remain undiscovered).

* In Section 3.2.4 please clarify how is the mask M_i generated for i^th speaker.

* In terms of speech separation background, there have been strong graphical model based approaches such as
Steven Rennie, John R. Hershey and Peder A. Olsen
"Single Channel Multi-talker Speech Recognition: Graphical Modeling Approaches,"
IEEE Signal Processing Magazine, Special issue on Graphical Models, November 2010.


**Time Spent Reviewing:**

15-20

---

> ### Author Response · Authors · 2021-08-10
> **To Reviewer UprN**
>
> **Q1: Is A-FRCNN used in both training and inference? Is it fair to call A-FRCNN an asynchronous update scheme for FRCNN?**
>
> A1：The A-FRCNN architecture is used in both training and inference. FRCNN (Figure 1a) is a recurrent model and it does not work on computers without unfolding it through time. The conventional unfolding scheme is the synchronous one as shown in Figure 1b, and we propose an asynchronous one as shown in Figure 1c. Note that FRCNN has 5 types of connections: (1) bottom-up adjacent connection; (2) bottom-up skip connection; (3) top-down adjacent connection; (4)  top-down skip connection; (5) lateral connection. If A-FRCNN has all these connections, then there is no problem to claim that it corresponds to an asynchronous updating scheme of FRCNN. However, in experiments we found that some skip connections were redundant. To save computation, we removed them (Lines 157-160, 165-166). In other words, A-FRCNN does not have all connections in FRCNN as illustrated in Figure 1a. But we think this small structure difference does not alter the main characteristics of the proposed method: asynchronous updating scheme. This name highlights the proposed connectivity pattern in contrast to the “synchronous” connectivity pattern in Figure 1b.
>
> **Q2: A new control model similar to S-FRCNN but without all skip-connections and skip connections in the time dimension.**
>
> A2: Thanks for this very interesting question. We considered a model similar to S-FRCNN but without all skip-connections in our submission to IJCAI-2021. We obtained results of “SI-SNRi: 11.43 dB, SDRi: 11.85 dB, Params: 6.40M, Inference Time: 0.51s”, with a 0.6 dB decrease in SI-SNRi compared to S-FRCNN. The results were much worse than A-FRCNN. We’ll add these results back to Section 5. Adding skip connections to the temporal dimension will be an interesting topic to explore.
>
> **Q3: Is there a method based on data-driven learning to determine the best design, or at least to guide this?**
>
> A3: It is possible to use data-driven learning such as the neural architecture search (NAS) techniques to determine the best design. In this manuscript we mainly argue that the update scheme, or connectivity patterns, in the FRCNNs can be greatly simplified and improved, and this can definitely serve as a starting point for future data-driven model structure search methods.
>
> **Q4: In Section 3.2.4 please clarify how is the mask M_i generated for i^th speaker.**
>
> A4: Here we assume that the number of sources to be separated in the mixture is known in advance, and we use a fully-connected layer with ReLU activation as the mask estimation layer (which is also the output layer) in the separator to estimate the multiplicative masks applied to the encoder outputs.
>
> **Q5: In terms of speech separation background, there have been strong graphical model based approaches.**
>
> A5: Thanks for reminding us about the graphical model-based approaches. We’ll add some discussion about those approaches including (Rennie, Hershey and Olsen, 2010) in the background part of our paper.

---

### Official Review · Reviewer_SsVq · 2021-07-17

**Rating:** 7
**Confidence:** 4

**Summary:**

This paper proposes a new speech separation architecture based on asynchrony fully recurrent CNN (A-FRCNN). The paper first describes FRCNN variants inspired by biological systems and discusses implementing such techniques for speech separation. Based on this discussion, the paper proposes a novel A-FRCNN speech separation technique. The experimental results show the effectiveness of the proposed method in two speech separation benchmarks (WHAM! and Librimix), achieving state-of-the-art performance compared with DPRNN. This new architecture can be applied to other similar problems, e.g., speech enhancement, and the contribution of this paper in terms of the technical novelty, experimental effectiveness, and potential wide applications is significant. My major concern about this paper is the clarity of the presentation. I could not fully understand the definition of the stage, bottom-up, and top-down operations until the actual speech separation discussions in Section 3.2. These concepts are not familiar, especially for speech separation researchers, and I recommend that the authors provide concrete examples of these concepts and elaborate the discussions.

Other comments:
1. Is it possible to use a transformer as a node instead of CNN? Recently, transformer-based processing becomes popular in speech separation.
2. Please discuss the effect of the reverberation. The real condition always has to deal with the reverberation.
3. Section 3.3 is a bit trivial, and it can be removed (or reduced). There are several important discussions in the appendix, and they can be put in the main document instead.

**Limitations And Societal Impact:**

The paper explicitly discusses the deficiency of the current techniques for more realistic scenarios.

**Main Review:**

originality:
The paper's concept of formulating speech separation problems with FRCNN is very novel for me.

quality:
Proposed A-FRCNN with several considerations of the implementation options and experimental validations make this paper very strong. Also, the paper has various experiments, including the ablation study, the careful comparisons with u-net, and the comparisons of the other methods with public benchmarks, which makes the experimental conclusions very valid.

clarity:
As I mentioned before, it is difficult to read the first part of the paper as it does not fully define/introduce the node, stage, bottom-up, top-down, lateral connections. I recommend the authors provide more concrete examples of them.

significance:
The paper's architecture and discussions can be applied to other machine learning problems (at least other speech processing problems, including speech enhancement). Thus, the impact of the paper would not be limited to speech separation and can have more general impacts in the machine learning community.

**Time Spent Reviewing:**

3

---

> ### Author Response · Authors · 2021-08-10
> **To Reviewer SsVq**
>
> **Q1: Definitions of the stage, bottom-up, and top-down operations are unclear until Sec 3.2.**
>
> A1：Thanks for your suggestion on the organization of the paper. We will define these concepts in the introduction section using concrete examples in Figure 1. Each stage corresponds to a group of neurons in a functional area in the sensory pathway (e.g., the inferior colliculus in the auditory pathway); both bottom-up and top-down connections can be made between adjacent stages and non-adjacent stages. Bottom-up and top-down connections can be implemented by downsampling (e.g., pooling or convolution with stride>1) and upsampling (e.g., interpolation or transposed convolution) operations, respectively.
>
> **Q2: Is it possible to use a transformer as a node instead of CNN?**
>
> A2: Thanks for pointing out this interesting point. We tried to use RNNs rather than CNNs in our pipeline where we adopted the dual-path RNN (DPRNN) architecture and incorporated our bottom-up and top-down connections (A-DPRNN, Sec. 2, Supplementary Materials), and we observed performance improvement compared to the baseline. Since the transformers are often used as a replacement to the RNNs in recent speech enhancement, separation, and recognition models, we think this can serve as a cue that transformer-based models can also benefit from our proposed design. Due to the time constraint, we did not further apply transformers as a direct replacement of the CNN nodes, however, we did extend the A-DPRNN architecture to the Sandglasset architecture [1&]. Sandglasset has a similar structure to DPRNN, and both have an intra-chunk and inter-chunk in every block. The major difference is that in Sandglasset, the BiLSTM in the inter-chunk is changed to a transformer. This modification improved SI-SNRi on WSJ0-2Mix dataset by 1.5 dB. We can transfer our method to Sandglasset in the same way as A-DPRNN. The resulting model is called A-Sandglasset and its structure is the same as A-DPRNN as illustrated in Figure S1b in the Supplementary Materials, and the only difference is that in the inter-chunk the BiLSTM is changed to a transformer. The original Sandglasset is too large to fit in our computing facility. We thus used a smaller version and applied our connection scheme. The Sandglasset hyperparameters are basically the same as those of DPRNN. Exceptionally, for the encoder/decoder module we set their kernel size to 16 and the number of channels to 256. The initial segment size was 64. The Sandglasset block input dimension was set to 128. The hidden layer dimension of BiLSTM and transformer were set to 128. The global Self-Attentive Network was set to be 8-head with a 0.1 dropout rate.
>
> The A-Sandglasset obtained 0.6 dB higher SI-SNRi value than the plain Sandglasset on the WSJ0-2Mix dataset. A one-second speech segment was used to calculate the GFlops, memory and inference time. The results are shown below.
>
> | Model | Params | GFlops | Forward/Backward GPU memory | Inference Time(CPU) | SI-SNRi|
> | --- | --- | --- | --- | --- | --- |
> | Sandglasset  | 2.3M | 4.08 | 0.16(GB)/0.17(GB)  | 0.28s | 17.3 dB|
> | A-Sandglasset  | 1.3M | 6.37 | 0.53(GB)/0.54(GB)  | 0.22s | 17.9 dB|
>
> [1&] Lam M W Y, Wang J, Su D, et al. Sandglasset: A Light Multi-Granularity Self-Attentive Network for Time-Domain Speech Separation, IEEE International Conference on Acoustics, Speech and Signal Processing (ICASSP), 2021, pp. 5759-5763.
>
>
> **Q3: Please discuss the effect of the reverberation. The real condition always has to deal with the reverberation.**
>
> A3: Thanks for this very important comment. In this manuscript we did not consider reverberation because the benchmark datasets we selected, for the sake of a fair and more straightforward performance comparison with other methods, mainly consider the anechoic case. We strongly agree that reverberation is the point that we always have to deal with in the real world, but given the page limit we are not able to include additional experiment results or discussions here. We will definitely evaluate our method under reverberation and noise in our next step.
>
> **Q4: Section 3.3 can be removed or reduced; several important discussions in the appendix can be put in the main document instead.**
>
> A4: Thanks for this valuable suggestion. We will rephrase Section 3.3 and move some experimental results and discussions in the Supplementary Materials to Section 5.

---

### Author Response · Authors · 2021-08-10
**To all reviewers**

We would like to thank all the reviewers for the helpful comments and suggestions, and we sincerely appreciate the time and efforts the reviewers put on this manuscript. Please find the point-to-point response to all the comments below.

---

### Decision · Program_Chairs · 2021-09-27

**Decision:**

Accept (Poster)

**Comment:**

This paper proposes to use a bio-inspired asynchronous fully recurrent convolutional neural network (A-FRCNN) for speech separation. In contrast to the conventional synchronous update, the authors argue that with asynchronous updates via the bottom-up, top-down and lateral connections in the network, the model can fuse information flows at various time scales for improved learning.  Experiments on WHAM! and Librimix demonstrate the effectiveness of the proposed approach.  The authors show that A-FRCNN can deliver high-performance results on these two datasets compared to some of the existing best performing models under the synchronous update mechanism.  While some of the results obtained by A-FRCNN are state of the art and some are slightly underperform,  A-FRCNN also offers high efficiency with fewer parameters.  The work is well motivated.  In the rebuttal, the authors also cleared numerous concerns raised by the reviewers.   Overall, this bio-inspired approach is novel and interesting.   I would recommend accept.  The authors should revise the paper accordingly based on suggestions in the reviews and discussion.